# Optimization of Process Conditions for Continuous Growth of CNTs on the Surface of Carbon Fibers

**Chengjuan Wang [1,2], Yanxiang Wang [1,2,*] and Shunsheng Su [2]**

[1] Key Laboratory for Liquid-Solid Structural Evolution and Processing of Materials (Ministry of Education), State Key Laboratory of Crystal Materials, Shandong University, Jinan 250061, China; wcj3519@163.com

[2] Carbon Fiber Engineering Research Center, School of Materials Science and Engineering, Shandong University, Jinan 250061, China; 202020478@mail.sdu.edu.cn

\* Correspondence: wyx079@sdu.edu.cn

**Abstract:** Grafting carbon nanotubes (CNTs) is one of the most commonly used methods for modifying carbon fiber surface, during which complex device is usually needed and the growth of CNTs is difficult to control. Herein, we provide an implementable and continuous chemical vapor deposition (CVD) process, by which the novel multiscale reinforcement of carbon nanotube (CNT)-grafted carbon fiber is prepared. After exploring the effects of the moving speed and growth atmosphere on the morphology and mechanical properties of carbon nanotubes/carbon fiber (CNTs/CF) reinforcement, the optimal CVD process conditions are determined. The results show that low moving speeds of carbon fibers passing through the reactor can prolong the growth time of CNTs, increasing the thickness and density of the CNTs layer. When the moving speed is 3 cm/min or 4 cm/min, the surface graphitization degree and tensile strength of CNTs/CF almost simultaneously reach the highest value. It is also found that $H_2$ in the growth atmosphere can inhibit the cracking of $C_2H_2$ and has a certain effect on prolonging the life of the catalyst. Meanwhile, the graphitization degree is promoted gradually with the increase in $H_2$ flow rate from 0 to 0.9 L/min, which is beneficial to CNTs/CF tensile properties.

**Keywords:** CNTs/CF; CVD; graphitization degree; growth atmosphere

## 1. Introduction

Carbon fiber has a series of excellent features such as high modulus, high strength, low density, high temperature resistance, and remarkable electrical conductivity. Therefore, it is often chosen as a reinforcing material to composite with a resin matrix, widely used in aerospace, transportation, new energy, building facilities, medical equipment, and military fields [1–3]. As the reinforcement, the performances of carbon fibers and the interfacial characteristics of the composites are important factors to determine whether the composites are excellent or not. However, during the carbonization treatment at high temperatures with inert gas, carbon fibers will meet the process of non-carbon element escaping and carbon enrichment, which will reduce the surface activity and surface tension of carbon fibers [4,5]. In addition, in order to ensure the tensile strength of carbon fibers, surface defects should be avoided as much as possible, so the surface of carbon fibers is smooth, and the specific surface area is not large. For these reasons, carbon fibers cannot be firmly compounded with the matrix material, which greatly reduces the properties of the composites. Thus, it is very important to modify the surface of carbon fiber to obtain proper surface roughness and activity.

At present, a great number of traditional surface modification methods of carbon fiber have been proposed to improve the wettability between carbon fibers and matrix materials, such as plasma modification, chemical oxidation, and electrochemical treatments [6–9]. Even though these methods do create the desired surface chemistry, the effects on the base fiber strength are adverse, resulting in poor mechanical properties. By

contrast, grafting carbon nanotubes (CNTs) is a more effective and moderate way [10,11]. The introduction and uniform distribution of CNTs on the surface of carbon fibers can significantly improve the mechanical properties of CNTs-grafted carbon fibers reinforced composites, and endow them with excellent electrical and thermal properties, expanding their application fields. Chemical vapor deposition (CVD) is usually used to prepare carbon nanotubes/carbon fiber (CNTs/CF) multiscale reinforcement [12–14], which can improve the adhesion between carbon fibers and the matrix [15–17], inhibit the crack growth and reduce the interfacial stress concentration [18–20]. An et al. [21] used a catalytic chemical vapor deposition (CCVD) method to grow vertically aligned CNTs arrays onto the carbon fiber fabric and found a nearly 110% increase in interfacial shear strength for the micro-droplet composite. Aerosol-assisted chemical vapor deposition was also developed to prepare carbon nanotube-hybridized carbon fiber [22]. Zhao et al. [23] reported an automatic fabrication of CNT fiber fabrics, which was achieved through knitting yarns composed of CNT fibers that are produced from a floating catalyst chemical vapor deposition. This kind of fabrics was proved to be flexible, lightweight, mechanically strong, and electrically and thermally conductive. Luo et al. [24] demonstrated the potential of carbon fibers modified by the vertical carbon nanotubes/polypyrrole composites for improving the electricity generation performances of the microbial fuel cells.

However, ordinary CVD techniques are not able to achieve continuous industrial production of CNTs/CF. Su et al. [25] provided a continuous method for grafting CNTs on the surface of carbon fibers by CVD, where the control of the morphology of CNTs/CF could be achieved by adding thiourea to the cobalt nitrate precursor. Yao et al. [26] developed a bimetallic catalyst for the growth of CNTs on the surface of carbon fibers and found that the catalytic efficiency is the highest when the atomic fraction of Fe is 1:1. Furthermore, the continuous method mentioned above would offer a promising technique for CNTs-grown carbon fibers online manufacture [27]. The growth of CNTs could also be achieved at ultra-low temperatures using the one-step method [28]. In the CVD process, in addition to the growth temperature [29] and catalyst system [30], the growth time and growth atmosphere [10] of CNTs have a great influence on the surface morphology and mechanical properties of CNTs/CF.

In this paper, this innovative continuous CVD process was used to grow CNTs on the surface of carbon fibers, with the mixture of $C_2H_2$ and $H_2$ as the growth atmosphere. By changing the moving speeds of carbon fibers and the volume ratio of $H_2$ to $C_2H_2$ in the mixed gas to adjust the growth time and atmosphere, a series of CNTs/CF multiscale reinforcements were synthesized under different process conditions. And the effects of process parameters such as carbon fiber moving speeds and mixed gas compositions on CNTs/CF multiscale reinforcements were explored and verified.

## 2. Materials and Methods

### 2.1. Experiment Method

The loading of the catalyst precursor and the continuous growth of CNTs were realized by using the equipment designed in the laboratory, as shown in Figure 1.

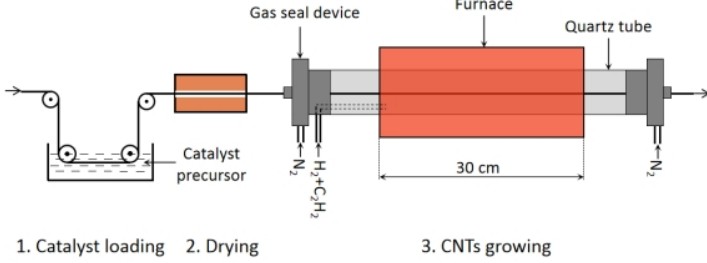

**Figure 1.** Schematic diagram of catalyst loading and carbon nanotubes (CNTs) growth equipment.

Polyacrylonitrile-based carbon fibers (T700-12000, Toray, Tokyo, Japan) were selected as original fibers for the growth of CNTs. The surface of desized carbon fiber was pretreated by electrochemical anodic oxidation (EAO). The mixed solution of cobalt nitrate hexahydrate ($Co(NO_3)_2 \cdot 6H_2O$, 0.05 mol/L, alcohol solution) and thiourea ($CH_4N_2S$, 0.02 mol/L, alcohol solution) was chosen as the precursor of catalyst. A tube furnace was used as the reactor, with a 30 cm long effective heating zone, and the diameter was 60 mm. In addition, the reducing agent for the catalyst was $H_2$, the carbon source was $C_2H_2$, and $N_2$ was adopted as the shielding gas. In the course of the experiment, the growth temperature was 600 °C. $H_2$ and $C_2H_2$ were inserted at the same time, after the furnace temperature reaching about 600 °C, to make the reduction of the catalyst and the growth of CNTs be carried out synchronously, which was called the one-step method, while the $N_2$ valve was opened from the beginning of the experiment.

Carbon fibers were driven by pulleys running at a directional and uniform speed. The moving speed of carbon fibers was adjusted by tuning the running speed of pulleys to control CNTs growth time, and the surface morphology and mechanical properties of CNTs/CF were studied. In addition, a variety of CNTs/CF was prepared by changing the flow rate of $H_2$ and $C_2H_2$ in the growth atmosphere. The flow rate of $C_2H_2$ remained 0.3 L/min, while $H_2$ flow rate increased as manifested in Table 1, influencing the volume ratio of $H_2$ to $C_2H_2$ and gas composition in the reactor.

**Table 1.** Different gas compositions in the growth atmosphere (The unit of flow rate is L/min).

| Gas | Flow Rate 1 | Flow Rate 2 | Flow Rate 3 | Flow Rate 4 | Flow Rate 5 |
|---|---|---|---|---|---|
| $H_2$ | 0 | 0.15 | 0.3 | 0.6 | 0.9 |
| $C_2H_2$ | 0.3 | 0.3 | 0.3 | 0.3 | 0.3 |

*2.2. Characterization*

The surface morphology of carbon fibers and multiscale reinforcements and the microstructure of CNTs was observed by field emission scanning electron microscope (SEM, SU-70) and high-resolution transmission electron microscope (HRTEM, JEM-2100). The degree of graphitization on the surface of multiscale reinforcements was characterized using the LabRAM-HR800 Raman spectrometer. The test condition was He-Ne light source, the wavelength was 633 nm, the resolution was 1 $cm^{-1}$, and the test range was 500–2000 $cm^{-1}$. The change of graphite microcrystals on the surface of carbon fiber was analyzed by X-ray diffraction (XRD, Rigaku D/max-RC).

The single-filament tensile strength of carbon fiber was tested according to ASTM D3822-07 standard. Samples were made according to Figure 2, and a single carbon fiber of appropriate length was fixed on the paper frame cut by a piece of mark paper with strong glue. After the glue was cured, two ends of the sample were clamped on the chuck of the XQ-1C fiber strength tester, and the paper pieces on both sides of the carbon fiber were cut with scissors to start the test. Stretching rate was 2 mm/min during the test. At least 40 effective samples were prepared for each condition, and the average value was taken. The single-filament tensile strength was calculated according to the following Formula.

$$\delta = \frac{4F}{\pi d^2} \tag{1}$$

In the Formula, $\delta$ is carbon fiber single-filament tensile strength (Pa), F means carbon fiber single-filament fracture load (N), $d$ is single carbon fiber diameter (m). The average diameter of carbon fiber took 7 μm.

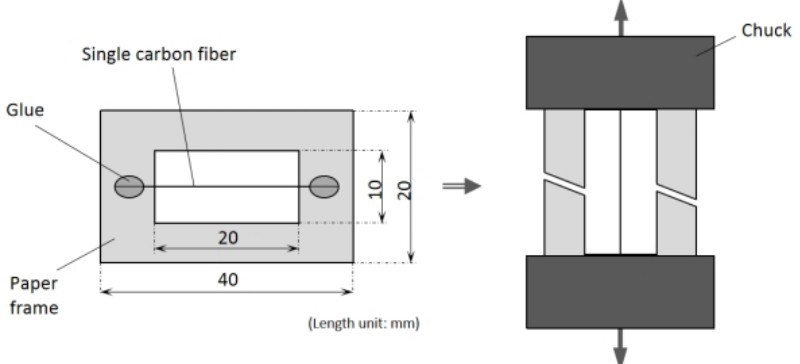

**Figure 2.** Schematic for the single fiber tensile testing.

## 3. Results and Discussion

### 3.1. Effect of Moving Speeds on Surface Morphology of Multiscale Reinforcements

Figure 3 shows the surface morphology of CNTs/CF multiscale reinforcements deposited at different moving speeds, respectively. It can be seen from the figure that due to the short deposition time, CNTs on the carbon fiber surface were sparse, short and uneven when the moving speed was 10 cm/min. When it came to 6 cm/min, the distribution of CNTs on the surface of carbon fibers became uniform, the length was longer, and CNTs between the two adjacent fibers were entangled with each other. At 4 cm/min or 3 cm/min, the growth time of CNTs was increased even more, making carbon atoms in the catalyst particles be able to diffuse and precipitate thoroughly on the surface of carbon fibers. Consequently, the length of CNTs became longer and CNTs were distributed more uniform and denser. As the growth time was prolonged to 15 min, a large amount of amorphous carbon was formed on the surface of carbon fibers, resulting in the hardening of the CNTs layer. In the box indicated by the yellow arrow in Figure 3e, the fracture surface of the carbon fibers could be meshed with each other, indicating that carbon fibers were initially connected by the entanglement of the CNTs layer, and then the brittle fracture occurred in the connecting area in the process of sample preparation. Plenty of amorphous carbon was filled in the gap of the CNTs layer in Figure 3f, which made CNTs bond with each other, leading to the binding of the CNTs layer and the increase in brittleness at the adjacent carbon fiber joints.

Figure 4 shows the changes of Raman curves and $R$ values of CNTs/CF multiscale reinforcements grown at various moving speeds. The two peaks of CNTs/CF exhibited in Figure 4 represent the disordered structure at about 1350 cm$^{-1}$ (D peak) and the ordered graphite structure at about 1600 cm$^{-1}$ (G peak), respectively. Usually, the higher the $R$ ($I_D/I_G$) value is, the worse the graphitization degree of the sample is [31]. It can be found that, from 10 cm/min to 3 cm/min, with the decrease of the moving speed, the growth time of CNTs became longer, and the $R$ value of samples went smaller gradually. This manifested that under the repairing effect of CNTs, the graphite microcrystalline structure on the fiber surface was perfected, the surface defects of carbon fibers decreased, bringing out an increase in the ordering of carbon atoms. As deposition time was extended further, the number of CNTs and the density of CNTs layer increased, the defects on the surface of carbon fibers due to EAO were repaired, which improved the degree of graphitization. When the moving speed was 4 cm/min or 3 cm/min, the $R$ value was similar, and the $R$ value decreased to the lowest level, illustrating that the degree of graphitization on the surface of the samples was the highest. The $R$ value suddenly increased to 0.791 in 2 cm/min, indicating that the samples had a pronounced reduction of graphitization degree. This was mainly due to the fact that the CVD process took too long, so a large number of catalyst particles lost their catalytic activity. Thus, active carbon atoms were excessive and fallen on the sample surface.

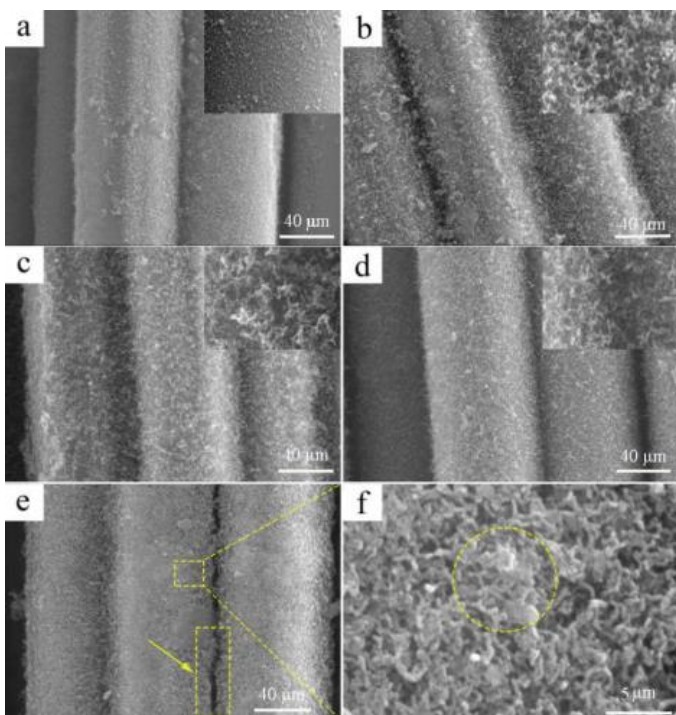

**Figure 3.** SEM micrographs of CNTs/carbon fiber (CF) multiscale reinforcements prepared at different moving speeds: (**a**) 10 cm/min; (**b**) 6 cm/min; (**c**) 4 cm/min; (**d**) 3 cm/min; (**e**) 2 cm/min and (**f**) the expanding image of (**e**). The flow rate of $H_2$ and $C_2H_2$ was 0.3 L/min.

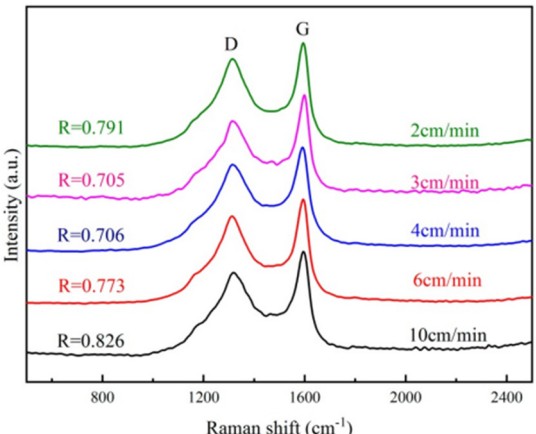

**Figure 4.** Raman spectra of CNTs/CF multiscale reinforcements prepared at different moving speeds. The flow rate of $H_2$ and $C_2H_2$ was 0.3 L/min.

Figure 5 manifests XRD curves of CNTs/CF multiscale reinforcements prepared at different moving speeds. It can be seen from the figure that each curve had an obvious diffraction peak at $2\theta = 25.3°$, corresponding to the (002) crystal plane of carbon fiber. The FWHM (full-width at half-maximum) value of the (002) crystal plane diffraction peak was analyzed to characterize the grain size of the samples, shown in Table 2. According to the Scherrer formula, the higher the value was, the smaller the grain size was, indicating that the (002) crystal plane was damaged more severely. With the increase in the moving speed, the FWHM value decreased at first and then raised. When the moving speed was 4 cm/min, the curve became the steepest and the FWHM value reached the minimum. The results showed that in the process of the carbon nanotubes' growth, activated carbon atoms continued to accumulate on the carbon fiber surface, and graphite microcrystal defects that

were oxidized and etched on the carbon fiber surface were repaired, which was helpful to improve the mechanical properties of carbon fiber.

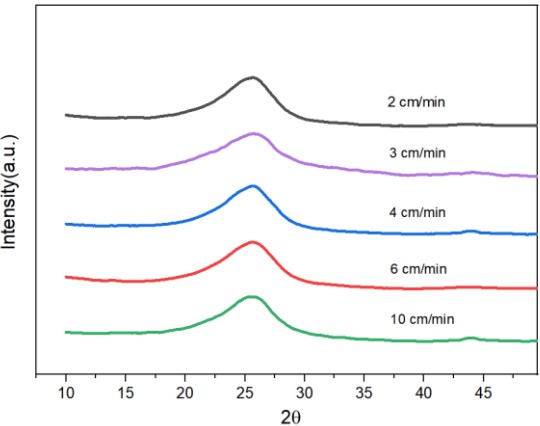

**Figure 5.** XRD curves of CNTs/CF multiscale reinforcements prepared at different moving speeds. The flow rate of $H_2$ and $C_2H_2$ was 0.3 L/min.

**Table 2.** FWHM values at $2\theta = 25.3°$.

| Sample | FWHM |
|---|---|
| 2 cm/min | 0.960 |
| 3 cm/min | 0.934 |
| 4 cm/min | 0.928 |
| 6 cm/min | 0.975 |
| 10 cm/min | 1.114 |

*3.2. Effect of Moving Speeds on Tensile Strength of Multiscale Reinforcements*

Figure 6 shows the effect of moving speeds on the single-filament tensile strength of CNTs/CF. It is worth noting that the mechanical properties of EAO treated carbon fibers was the poorest compared to other samples. This was mainly because the surface of desized carbon fiber was etched and damaged during the EAO treatment, contributing to the decrease of tensile strength, lower than that of the desized carbon fibers. At 4 cm/min, tensile strength reached the maximum value of 4.46 GPa, which was about 5.4% higher than that of the desized carbon fibers and 8.8% higher than that of the electrochemically treated carbon fibers. This was because, with the prolonged CVD process time, the defects on the surface of carbon fibers were repaired and made up gradually by a great number of newly deposited CNTs of the highest quality. Thus, the strength of the fibers was enhanced, presenting a more effective reinforcing effect of the CVD process, which corresponded to the result of SEM and Raman images given above. However, when carbon fibers moved quicker than 4 cm/min, the tensile strength of the samples was decreased correspondingly with the increase in moving speeds. Because of the short deposition time, CNTs/CF synthesized at 10 cm/min with unsatisfactory surface morphology presented very low mechanical properties, almost equivalent to the EAO treated fiber. As the moving speed went down to 3 cm/min, the tensile strength of the samples was reduced slightly compared to the highest strength. While the continuous carbon fibers passed through the reactor at the speed of 2 cm/min, the value was about 5% lower than the biggest one. This phenomenon was mainly related to the life of the catalyst, which led to the rising *R* value in Figure 4. Catalyst particles were deactivated one after another when the CVD process took a long time, and the deactivated catalyst particles would no longer facilitate the growth of CNTs. As a result, defects could not be obviously repaired, so the tensile strength of multiscale reinforcements was no longer improved. If the CVD time continued to be added at this time, a great quantity of amorphous carbon would be deposited on the surface of the

carbon fibers, which not only could not cause an increase in the tensile strength of the fibers but also adversely affect the preparation of samples.

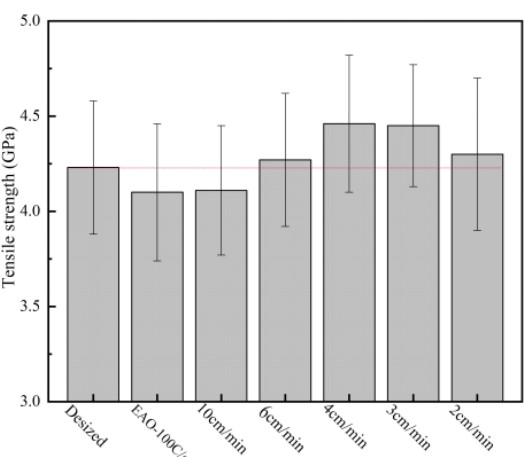

**Figure 6.** Single-filament tensile strength test of CNTs/CF multiscale reinforcements prepared at different moving speeds. The flow rate of $H_2$ and $C_2H_2$ was 0.3 L/min.

### 3.3. Effect of Growth Atmosphere on Morphology of Multiscale Reinforcements

In addition, different growth atmosphere compositions also affected the morphology of CNTs. In the tube reactor kept at 600 °C, the carbon source ($C_2H_2$) was pyrolyzed at high temperature and under the action of the catalyst to form active carbon atoms and release hydrogen. Then the catalyst converted active carbon atoms into CNTs. In Figure 7a–e, the flow rate of $C_2H_2$ remained the same at 0.3 L/min, and the flow rate of $H_2$ increased from 0 to 0.9 L/min. It was obvious that under the condition of no $H_2$, there were few CNTs grown on the surface of carbon fibers. The precursor of catalyst decomposed at high temperature to form metal oxides with low catalytic activity, giving rise to insufficient reduction of catalyst. Moreover, the pyrolysis reaction of the carbon source was not restrained, so a large number of active carbon atoms would be produced and transformed into pyrolytic carbon wrapped on the surface of the catalyst in a short time, resulting in the deactivation of the catalyst. These two reasons affected the normal growth of CNTs, leading to a small quantity of CNTs and nonuniform distribution [32].

After a small amount of $H_2$ was injected into the growth atmosphere, the reduction of some catalysts became sufficient, and the life of the catalyst was prolonged, which was beneficial to the growth of CNTs. Therefore, the number of CNTs raised significantly, and the introduction of $H_2$ inhibited the cracking of carbon source to a certain extent [33,34]. When the $H_2$ flow rate increased to 0.3 L/min, the CNTs' deposition began to be uniform. And the length and number of CNTs got a pronounced increase as the $H_2$ flow rate reached 0.9 L/min. Additionally, CNTs were entangled with each other, agglomerated or connected into sheets.

As shown in Figure 8, with the addition of $H_2$ content, the *R* value became smaller step by step, demonstrating that the degree of graphitization on the surface of the samples was promoted. This indicated that the regularity of carbon atoms on the CNTs got an increase and the number of impurities such as amorphous carbon reduced. By adjusting the content of $H_2$ and making use of the inhibitory effect of $H_2$ on the cracking of carbon source, the surface morphology and graphitization degree of CNTs/CF multiscale reinforcements could be controlled [35].

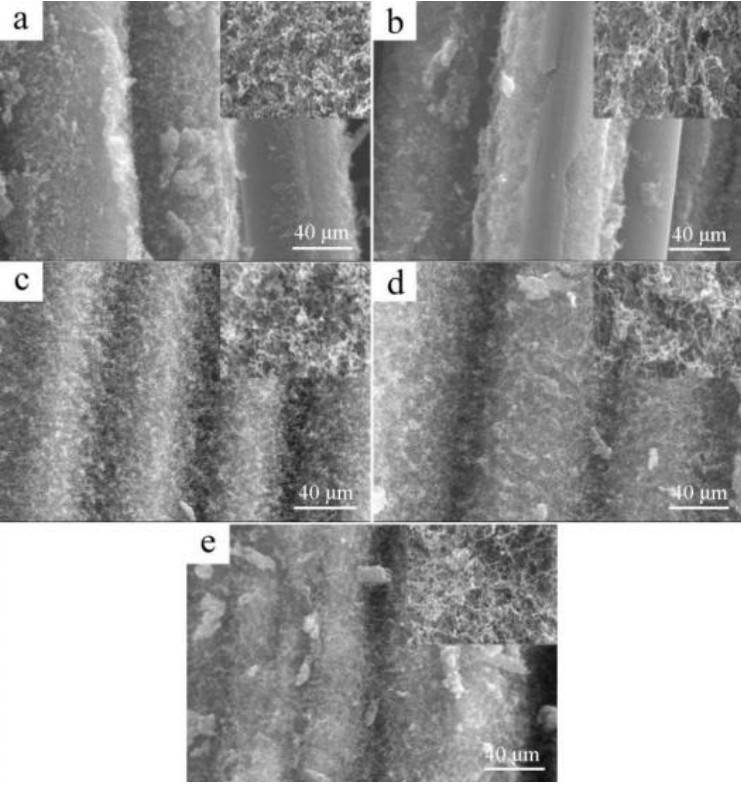

**Figure 7.** SEM micrographs of CNTs/CF multiscale reinforcements prepared under different growth atmospheres, the flow rate of $H_2$ to $C_2H_2$: (**a**) 0:0.3; (**b**) 0.15:0.3; (**c**) 0.3:0.3; (**d**)0.6: 0.3 and (**e**) 0.9:0.3. The moving speed of carbon fibers was 4 cm/min.

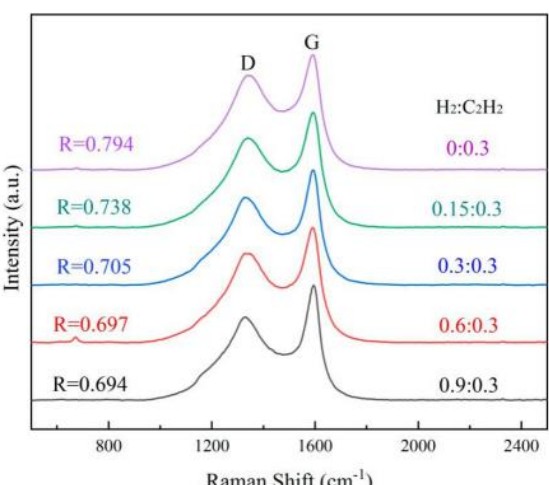

**Figure 8.** Raman spectra of CNTs/CF multiscale reinforcements prepared under different growth atmospheres. The moving speed of carbon fibers was 4 cm/min.

### 3.4. CNTs Microstructure Analysis

Figure 9 shows the detailed microstructure of CNTs on the surface of CNTs/CF prepared when the moving speed was 4 cm/min and $H_2$:$C_2H_2$ = 0.3:0.3. From SEM images and Figure 9, we can obtain that the diameter of CNTs with excellent character was concentrated around 9–12 nm, while the inner diameter was around 3.5–4.5 nm, and the average length of CNTs was about 40 nm. As seen in Figure 9, the CNTs grown under these two growth conditions had a large aspect ratio, an obvious hollow structure and tubular morphology, and the distribution was uniform and dense. From Figure 9b,d, it could be clearly observed that the thickness of CNTs tubes was even, the inner diameter

and the outer diameter were coaxial. Additionally, the graphite sheet on the tube wall, or the lattice fringes of the CNT, basically aligned parallel to the CNTs' axis, indicating the high crystallinity. This indicated that the CNTs grafted under these two conditions had fewer wall defects, and the degree of graphitization of CNTs was the largest [36]. What is more, the catalyst particles were located at the end of the CNTs far away from the surface of carbon fibers and would fall off from the top of CNTs during the sample preparation process, resulting in an open state at the top of the CNTs. This type of structure could be found easily in Figure 9.

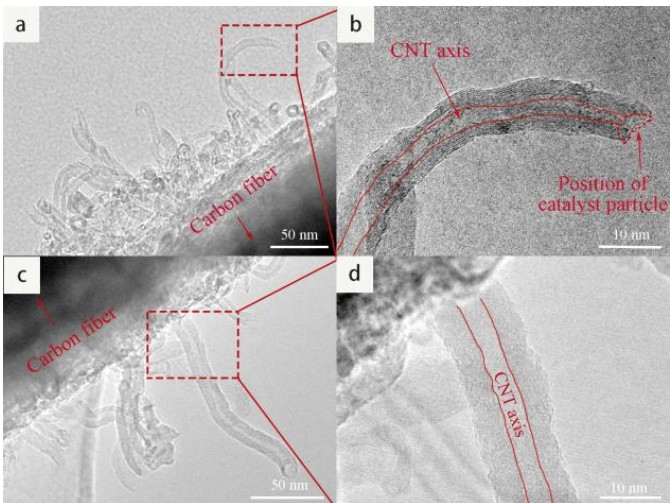

**Figure 9.** HRTEM images of CNTs grafted on the samples: 4 cm/min and $H_2$:$C_2H_2$ = 0.3:0.3.

### 3.5. Effect of Growth Atmosphere on Tensile Strength of Multiscale Reinforcements

Figure 10 reflects the effect of growth atmosphere on the mechanical properties of CNTs/CF. In the absence of $H_2$, the single-filament tensile strength of the sample was 4.33 GPa, which was about 2.8% higher than that of the desized carbon fibers. After the injection of $H_2$, the tensile strength was increased with the addition of $H_2$ flow rate. As it went up to 0.3 L/min, the tensile strength was 4.41 GPa, reaching the maximum value of the system, which was about 4.3% higher than that of the desized carbon fibers. When the rate was added to 0.6 L/min, the tensile strength did not make a change obviously. But the tensile strength was reduced to 4.38 GPa when it came to 0.9 L/min. $H_2$ could restrain the cracking of the carbon source and reduce the production of amorphous carbon, so injecting an appropriate amount of $H_2$ in the growth atmosphere could improve the efficiency of the catalyst, which was beneficial to the reinforcement of the CVD process. However, when the flow rate of $H_2$ was too large, the inhibition on the cracking of carbon source was too strong, and the number of active carbon atoms decreased in the early stage of CNTs growth, which would make the surface of carbon fiber more seriously etched, and finally led to the decrease of single-filament tensile strength of samples. Additionally, the high crystallinity of CNTs shown in TEM images had a positive effect on improving CNTs/CF mechanical properties. That was why the tensile strength of carbon fibers increased significantly after the deposition of CNTs.

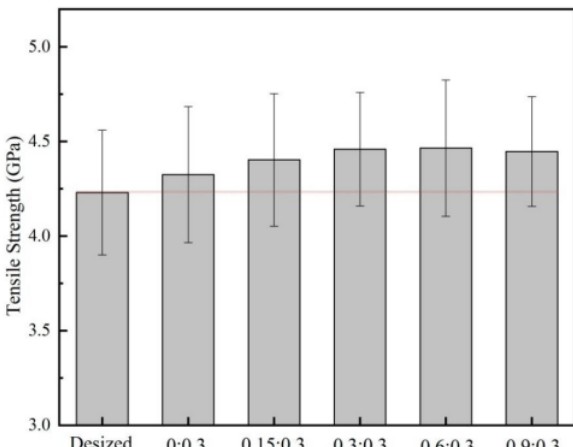

**Figure 10.** Single-filament tensile strength test of CNTs/CF multiscale reinforcements prepared under different growth atmospheres. The moving speed of carbon fibers was 4 cm/min.

## 4. Conclusions

In conclusion, by tuning the moving speeds of carbon fibers and the ratio of $C_2H_2$ to $H_2$ in the growth atmosphere, effects of deposition time and growth atmosphere on the surface morphology and tensile strength of CNTs/CF multiscale reinforcements were systematically investigated. With the decrease of moving speeds, the thickness and density of the CNTs layer gradually increased, and the degree of graphitization of CNTs/CF multiscale reinforcements was improved at first and then had a reduction. When the moving speed was 3 or 4 cm/min, the distribution of CNTs was ideal, and the degree of graphitization on the surface of the sample was the highest. In addition to reducing catalyst, $H_2$ had a certain inhibitory effect on carbon source pyrolysis. Adding the proportion of $H_2$ in the growth atmosphere could reduce the production of amorphous carbon and prevent the deactivation of the catalyst, which was beneficial to the growth of CNTs, improving the surface graphitization and crystal structure of CNTs/CF multiscale reinforcements.

**Author Contributions:** Conceptualization, C.W. and Y.W.; Data curation, C.W.; Investigation, C.W. and Y.W.; Methodology, C.W.; Project administration, Y.W.; Software, S.S.; Supervision, Y.W.; Validation, C.W.; Visualization, C.W.; Writing—original draft, C.W. and S.S.; Writing—review and editing, C.W., Y.W. and S.S. All authors have read and agreed to the published version of the manuscript.

**Funding:** This research was funded by Natural Science Foundation in Shandong Province (ZR2020ME039, ZR2020ME134) and National Natural Science Foundation of China (51773110).

**Institutional Review Board Statement:** Not applicable.

**Informed Consent Statement:** Not applicable.

**Acknowledgments:** The authors thank the editor and the anonymous reviewers for their valuable comments on this manuscript. The authors also acknowledge the support of technical staff for assisting in preparing samples and analyzing them.

**Conflicts of Interest:** The authors declare no conflict of interest.

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
