# Peer review of "Optimization of Process Conditions for Continuous Growth of CNTs on the Surface of Carbon Fibers"

_jcs, doi:10.3390/jcs5040111_

Round 1
Reviewer 1 Report
The paper is well written and discussion is supported by results.
There are only two errors in rows 90-92:
"And the mixed solution of cobalt nitrate hexa-90 hydrate (Co(NO3)2·6H2O, 0.05 mol/L, alcohol solution) and thiourea (CH4N2S, 0.02 mol/L, 91 alcohol solution) was confected as the precursor of catalys."
I suggest to publish that paper.
Reviewer 2 Report
In this manuscript, the authors show a screening of conditions for the synthesis of CNTs grafted to a CF by a continuous CVD process. The CNTs could improve the performance of CFs as fillers in polymeric composites. The manuscript can be quite easily understood, and develops novel ideas in the field of composite carbon fillers. I recommend this manuscript for publication after minor revision:
- The authors suggest the CNT/CF structure can be used as an improved filler for polymers. In that case, a good transfer of the mechanical property from CNTs to CF through the junction (at the CNT base) would be needed. Could the authors include a comment concerning the mechanical transfer in the junction?
- Abstract, line 15. The way of “moving speed” cannot be understood by the reader without going to the main text. Could the authors clarify the experiment mode? (the CF yarn seems to be continuously propelled by pulleys).
- Introduction, line 39: “Thus, making CF gain…” This sentence seems contradictory and incomplete in the paragraph. Why the “better wettability” is needed?
- Section 2.1. Some experimental details are missing: the reactor diameter; gas flow rates and compositions could be summarized in a table; are H2 and C2H2 inserted at the same time or sequentially?
- Characterization. Techniques such as X-ray diffraction should be considered to consistently discuss the “graphitization”. However, Raman G and D bands are not directly related to the stacking of layers, but with log-range order inside the layers.
- From SEM and TEM images, diameter and length distributions (aspect ratio) should be determined.
- Captions of Figures 3, 4, 5, 6, 7, … are incomplete. Please, include the gas mixture, the growth atmosphere, and the moving speed in each case.
- Section 3.3. The role of h2 is well-known in the literature, and should be cited, e.g.: Carbon 47 (2009) 998-1004; Carbon 47 (2009) 1989-2001; etc.
Reviewer 3 Report
The reviewed article presents the results of the research concerning the selection of conditions for the growth process of carbon nanotubes on carbon fibers surface.
As the main process parameters, the authors optimized the CF moving speed inside the furnace and the composition of the reaction atmosphere. Despite the well-known research issue, the article is interesting. After reading the manuscript, I have a few comments and questions:
Was one single fiber or a bundle of fibers drawn in the reaction atmosphere? If the fiber bundle was each surface equally populated with CNTs?
How was the diameter of the modified fibers measured for strength calculations?
What was the surface of the fibers after EDA, please add an SEM image. How did the fiber diameter change after this process?
What is, according to the authors, the mechanism of strengthening the tensile strength of fibers by growing CNT on their surface.
What's in photo 6e in the form of plate structures? are they CNT aggregates or amorphous carbon?
Please check the correctness of plotting the confidence intervals on the graphs - they all have almost the same range, and this is rare.
The yellow color in Fig. 8 is hardly readable.
Round 2
Reviewer 3 Report
The authors made a comprehensive explanation of my doubts and took into account the proposed suggestions for changes in the manuscript. In its current form, the revised work presents a significant scientific value, therefore I agree with the authors' request and recommend the work for publication in the J. Compos. Sci.